# Effect of Electron-Beam Irradiation on Microbiological Safety, Nutritional Quality, and Structural Characteristics of Meat

**DOI:** 10.3390/foods14091460

**Published:** 2025-04-23

**Authors:** Duman Orynbekov, Kumarbek Amirkhanov, Zhanar Kalibekkyzy, Nazerke Muslimova, Gulnur Nurymkhan, Almagul Nurgazezova, Samat Kassymov, Amirzhan Kassenov, Aigul Maizhanova, Botakoz Kulushtayeva, Zhanibek Yessimbekov

**Affiliations:** 1Faculty of Engineering and Technology, Shakarim University, 20A Glinka Str., Semey 071412, Kazakhstan; duman_r@mail.ru (D.O.); aspirant57@mail.ru (K.A.); zhanar_moldabaeva@mail.ru (Z.K.); muslimova.n.r@mail.ru (N.M.); gulnu-n@mail.ru (G.N.); almanya1975@mail.ru (A.N.); samat-kasymov@mail.ru (S.K.); fquekm2710@mail.ru (A.M.); kulushtaeva_89@mail.ru (B.K.); 2Department of Technology of Food and Processing Industries, S. Seifullin Kazakh Agro-Technical Research University, 62 Zhenis Ave., Astana 010011, Kazakhstan; amirzhan-1@mail.ru; 3Kazakh Research Institute of Processing and Food Industry (Semey Branch), Semey 071410, Kazakhstan

**Keywords:** meat irradiation, electron-beam, microbial safety, textural properties, amino acid composition, mineral content

## Abstract

Foodborne pathogens remain a significant global challenge, contributing to widespread illness and considerable food losses. This study investigates the effects of electron-beam irradiation on beef quality and safety using a pulsed high-frequency linear accelerator (ILU-10). Meat samples were subjected to irradiation at doses of 3, 6, and 9 kGy, with non-irradiated samples serving as controls. The research focused on evaluating microbial reduction, alterations in textural properties, and changes in nutritional components including amino acids, vitamins, and mineral content. Microbiological analysis demonstrated a dose-dependent reduction in total viable counts, with a decrease from 300 CFU/g in controls to 100 CFU/g at 3 and 6 kGy and complete microbial inactivation at 9 kGy. Scanning electron microscopy revealed disruption in myofibrillar structure, with increased interstitial spacing. Chemical analyses indicated a dose-dependent decline in total amino acid content and variable responses among individual amino acids, suggesting irradiation-induced protein fragmentation and oxidation. The findings suggest that, when optimized, irradiation can substantially improve meat safety while maintaining acceptable nutritional and sensory quality.

## 1. Introduction

Food spoilage and foodborne illnesses remain critical challenges to global public health and food security. Approximately 600 million individuals fall ill annually, resulting in 420,000 deaths worldwide due to foodborne pathogens [1]. Furthermore, around 1.3 billion tons or nearly 30% of global food production is lost each year due to microbial spoilage, substantially impacting economic resources and food availability [2,3]. The meat industry, in particular, faces considerable safety risks due to microbial pathogens such as *Salmonella*, *Escherichia coli*, *Listeria monocytogenes*, and other spoilage organisms, which frequently contaminate meat at various stages—from processing plants to retail shelves [4,5].

Traditional preservation methods—such as chemical additives, thermal processing, and refrigeration—while effective to a degree, often compromise the nutritional quality and sensory attributes of meat or require high energy inputs. Against this backdrop, food irradiation has emerged as a promising non-thermal technology that enhances food safety, extends shelf life, and minimizes nutritional losses [6].

Food irradiation involves exposing food products to controlled doses of ionizing radiation (gamma rays, X-rays, or electron beams) to eliminate or significantly reduce microbial contamination [7]. This method has been endorsed by leading regulatory authorities such as the U.S. Food and Drug Administration (FDA), the World Health Organization (WHO), and the U.S. Department of Agriculture (USDA), which recognize irradiation as a safe and efficient means to ensure food safety. For example, the FDA permits doses up to 4.5 kGy for uncooked, chilled red meats and up to 7 kGy for frozen products [8]. Similarly, the WHO has consistently declared that irradiation at doses up to 10 kGy does not pose any toxicological hazard and does not compromise the nutritional quality of food, provided that the process is carefully controlled [9].

Despite these advancements and regulatory endorsements, consumer acceptance remains a challenge. Many consumers are hesitant due to misconceptions linking irradiation with radioactivity or potential chemical alterations in food. Contrary to these beliefs, extensive research confirms that irradiation neither makes food radioactive nor significantly alters nutritional or sensory qualities beyond the levels typically observed in conventional cooking methods [10]. Notably, irradiated foods have been consumed safely for decades in specialized contexts, such as NASA space missions, further underscoring their safety profile [11].

Scientifically, irradiation acts primarily by breaking microbial DNA or disrupting cellular integrity through oxidative processes induced by free radicals. This leads to a dramatic reduction in pathogenic and spoilage microorganisms without the need for elevated temperatures. Thus, irradiation effectively preserves the fresh qualities of meat products and reduces reliance on chemical preservatives [12]. However, it is also true that the irradiation process may induce certain chemical reactions, such as lipid oxidation or formation of volatile sulfur compounds, which may negatively influence sensory properties, particularly at higher doses (above 10 kGy) [13].

Scientific studies have demonstrated that ionizing radiation can accelerate the oxidation of fats [14,15] and result in the formation of off-odors [16,17]. These adverse effects are primarily attributed to the formation of free radicals during irradiation, which initiates a cascade of chemical reactions. For example, the generation of free radicals may lead to the production of volatile sulfur compounds or carbon monoxide. Moreover, at higher doses, the cellular structure of the meat is disrupted, proteins undergo structural changes, and enzymatic activities are inhibited, resulting in a soft texture accompanied by an unpleasant flavor [18,19].

In the work of Donskova and Belyaev [20], minced poultry meat irradiated at a dose of 30 kGy (using ^60^Co) exhibited significant deterioration in organoleptic properties, characterized by a gray color, a burning odor, and an overall decline in visual quality. In contrast, samples treated at a lower dose of 12 kGy maintained acceptable organoleptic characteristics, suggesting that there exists an optimal irradiation window where microbial safety is achieved without compromising quality. Consequently, optimal dose selection is crucial to balancing microbial safety with sensory and nutritional quality.

Consumer acceptance remains one of the most significant barriers to widespread adoption. Studies have shown that public understanding significantly impacts acceptance; providing scientifically accurate information and clear labeling (e.g., the Radura symbol) can significantly improve consumer perception and acceptance [21]. Thus, improved communication strategies about irradiation’s safety, benefits, and process transparency are vital for enhancing public trust and market penetration [22].

Food irradiation represents a robust, scientifically validated strategy to enhance meat safety and shelf life, mitigating global food spoilage issues. The application of ionizing radiation presents a promising solution to the persistent challenge of foodborne pathogens and spoilage in meat products. However, the selection of an appropriate irradiation dose is critical, as it directly influences both microbial safety and the preservation of nutritional and sensory qualities.

This study offers a novel contribution by focusing on meat irradiation specifically for Kazakhstan—a region with a rapidly growing meat production sector yet limited research on advanced preservation techniques. It delivers specific insights on the optimal irradiation dosage that eliminates pathogens while preserving nutritional quality, essential for developing regional food safety protocols that can enhance shelf life and ensure safety during logistics and production, in strict accordance with international and Kazakhstan’s regulatory authorities.

This research aims to study the dose-dependent effects of irradiation on meat, focusing on its impact on various quality parameters, including microbiology, amino acid composition, mineral profile, and textural properties. By optimizing irradiation protocols, the goal is to achieve a balance between ensuring food safety and maintaining the inherent quality of meat products, thereby addressing both consumer and industry concerns.

## 2. Materials and Methods

### 2.1. Sample

The meat samples were obtained immediately after the slaughtering of cows (2–2.5 years old) from the peasant farm “Mukinov” of Ernazar village, Beskaragai district, Abai region (Kazakhstan). The meat obtained after slaughter was cooled in refrigerated chambers at 0–(+2) °C for 24 h. After cooling, meat was vacuum-packed and frozen to −18 °C.

### 2.2. Radiation Treatment of Meat Samples

Electron-beam irradiation of meat samples was performed using a pulsed high-frequency linear electron accelerator ILU-10. The ILU-10 accelerator comprises the following main components (Figure 1): vacuum tank, copper toroidal resonator, focusing magnetic lens, magnetic discharge pumps for vacuum creation, electron injector (triode electron gun), exhaust device, communication loop, vacuum capacitor, and two radio-frequency (RF) generators operating at 118 MHz [23].

The ILU-10 accelerator operates within an electron energy range of 2.5 to 5 MeV, achieving electron-beam power of up to 50 kW. At an energy of 5 MeV, the ILU-10 delivers pulsed beam currents up to 350 mA with an average current of 4.3 mA. Electron-beam scanning occurs across an 800 mm width with a conveyor speed adjustable from 2 to 8 cm/s. The accelerator’s electron release window measures 980 mm in length by 80 mm in width and is covered by a thin titanium foil (thickness 50 µm).

### 2.3. Experimental Procedure

Beef samples (*Longissimus dorsi* muscle) weighing approximately 150–200 g each were individually sealed under vacuum conditions (40–60 mm Hg) in polyethylene bags measuring 20 × 25 cm (Figure 2). The average chemical composition was protein 25%, fat 15%, moisture 59%.

Electron-beam treatment was conducted sequentially from two opposite sides of each sample to ensure uniform dose distribution throughout the product volume. Samples were irradiated at three predetermined doses based on standard food processing applications: 3 kGy, 6 kGy, and 9 kGy. The exposure duration was set to 240 s for each treatment session.

The choice of these particular doses (3, 6, and 9 kGy) was guided by regulatory standards and previous scientific literature [7,8,9,15,16], which identifies 3 kGy as a moderate dose effective for significantly reducing pathogens; 6 kGy as intermediate (addressing a wider spectrum of microorganisms including resistant vegetative cells); and 9 kGy as a higher but still practically relevant dose—approaching the upper limit recommended by international authorities for broader microbial inactivation, including spores and viruses.

### 2.4. Irradiation Conditions and Parameters

During irradiation processing, strict control of the absorbed dose and its uniform distribution throughout the sample volume was maintained. Key parameters influencing dose distribution and uniformity, including electron energy, geometry and density of samples, conveyor speed, and electron-beam scanning width, were carefully monitored. Detailed irradiation parameters utilized for the experimental trials are summarized in Table 1 and Table 2 below:

The irradiation procedure and conditions were specifically chosen to ensure a precise and uniform treatment of meat samples, aiming to achieve reliable microbiological safety improvements and assess any physicochemical and sensory modifications in the treated meat.

### 2.5. Determination of Microbiological Indicators

After undergoing irradiation, the samples were transported immediately with crushed ice and then maintained at a temperature of approximately 4 ± 1 °C throughout the microbial testing period. This refrigerated storage is critical to accurately assess the efficacy of the irradiation process in controlling microbial populations.

Microbiological indicators were determined through a comprehensive analytical process involving multiple stages of sample preparation and quantification. Initially, a 10 g sample was weighed and prepared in a series of tenfold dilutions using a sterile saline solution. Total viable count (TVC), total coliform count (TCC), and yeasts and molds were assessed using Petritest™ (Saratov, Russia) substrates with standardized incubation conditions: TVC and TCC at 36 ± 1 °C for 12–24 h and yeasts/molds at 24 ± 1 °C for 24 h (preliminary) and 120 h (final). For each analysis, 0.2 cm^3^ of appropriate dilution was pipetted onto the substrate and evenly distributed, with colonies counted within specific ranges (15–300 for TVC, 15–150 for yeasts, 5–50 for molds). Results were calculated by multiplying colony counts by the respective dilution factors and normalized to colony-forming units (CFUs) per 1 cm^3^ of sample, ensuring precise and reproducible microbiological quantification [24].

For quantitative determination of Clostridium perfringens in meat products, we used the method based on ISO 7937:2004 [25], which provides for homogenization of 25 g of ground sample in 225 mL of sterile diluent followed by preparation of tenfold dilutions. Then, 1 mL of each dilution was seeded into Petri dishes with sulfite-cycloserine agar (SC-agar), filled with molten medium at 44–47 °C, and incubated under anaerobic conditions at 37 °C for 20 ± 2 h. After incubation, typical black colonies were counted in dishes with no more than 150 colonies and confirmed by microscopic examination for Gram-positive bacilli, lecithinase activity test on yolk medium, and ability to ferment lactose with gas formation in lacto-sulfite medium.

### 2.6. Determination of Protein Content

Protein content in meat samples was determined using the standard Kjeldahl method, a comprehensive procedure for nitrogen quantification. A representative 1–2 g meat sample was digested with concentrated sulfuric acid, potassium sulfate, and a copper sulfate catalyst at 350–400 °C until complete mineralization, converting organic nitrogen to ammonium sulfate. After cooling and dilution, the sample was neutralized with 40% sodium hydroxide to release ammonia, which was then distilled into a boric acid solution [26]. The resulting ammonium borate was titrated with standard sulfuric acid, and the nitrogen content was calculated based on the acid volume used. Total protein was estimated by multiplying the nitrogen content by the standard conversion factor of 6.25, with results expressed as a percentage of sample weight.

The mass fraction of protein is calculated by Formula (1):(1)$$X=\frac{0.0014\times (V_{1}-V_{2})\times K\times 100}{m}\cdot 6.25$$
where 0.0014 is the amount of nitrogen equivalent to 0.1 mol/dm^3^ of hydrochloric acid solution, g;

*V*_1_ is the volume of 0.1 mol/dm^3^ of hydrochloric acid solution or the volume of 0.05 mol/dm^3^ used for titration of the test sample, cm^3^;

*V*_2_ is the volume of 0.1 mol/dm^3^ of hydrochloric acid solution or the volume of 0.05 mol/dm^3^ used for titration of the control sample, cm^3^;

*K* is the correction factor to the nominal concentration of hydrochloric acid solution;

100 is the percentage conversion factor;

*m* is the mass of the sample, g;

6.25 is the protein conversion factor.

### 2.7. Determination of Fat Content

Fat content in meat samples was determined using the standard GOST 23042-2015 method with Soxhlet extraction [27]. A representative 2–5 g meat sample was initially dried to a constant weight at 105 °C then placed in a porous thimble within the Soxhlet apparatus for fat extraction using petroleum ether. The extraction process involved 6–10 cycles over 4–6 h, with solvent repeatedly percolating through the sample to dissolve and accumulate fat. Following extraction, the petroleum ether was completely evaporated using a rotary evaporator, and the remaining fat was dried to constant weight at 105 °C.

The mass fraction of fat X as a % is calculated according to the Formula (2):(2)$$X=\frac{\left(m_{2}-m_{1}\right)\times 100}{m}$$
where *m*_2_ is the mass of the extraction flask with fat, g;

*m*_1_ is mass of the extraction flask, g;

100 is the percentage conversion factor;

*m* is the mass of the analyzed sample, g.

### 2.8. Determination of Water Content

A representative 5–10 g meat sample was mixed with pre-dried sand in a pre-weighed drying dish to enhance moisture removal and increase surface area. The sample was dried in an oven at 103 ± 2 °C for 2–4 h, with periodic removal and weighing to ensure constant weight. After cooling in a desiccator to prevent moisture reabsorption, the final weight was determined, with repetitive drying cycles performed until mass stabilization (weight difference < 0.001 g). Moisture content was calculated as a percentage by comparing the initial and final sample weights, providing a precise quantification of water content in the meat sample.

### 2.9. Determination of Ash Content

A representative 2–5 g meat sample was placed in a pre-weighed crucible and subjected to a gradual temperature increase in a muffle furnace, progressing to 550 °C over 5–6 h to systematically dry, char, and ash the sample. The ashing process continued until a grayish-white residue remained, typically requiring 2–4 h of complete combustion. After cooling in a desiccator to prevent moisture absorption, the crucible with ash residue was weighed, with repeated heating and cooling performed to ensure constant mass (weight difference < 0.001 g).

The mass fraction of total ash, *X* %, is calculated with Formula (3):(3)$$X=\frac{(m_{2}-m_{0})\times 100}{\left(m_{1}-m_{0}\right)}$$
where *m*_2_ is the mass of the cup with ash, g;

*m*_0_ is the cup weight, g;

*m*_1_ is the mass of the cup with the test sample, g.

### 2.10. Determination of Shear Force

Shear force in meat samples was evaluated using a structurometer (Radius Company, St. Petersburg, Russia) to measure the mechanical resistance of the meat [28]. Samples were prepared either using a specialized rotating tubular knife to create a 10 mm diameter cylindrical specimen or manually cut into a 20 mm × 20 mm square shape. The prepared sample was positioned on the structurometer’s flat table, where a cutting force was applied to determine the shear force. The force required to cut through the sample was directly recorded from the structurometer’s scoreboard, providing a quantitative measure of the meat’s mechanical properties. The shear stress was determined using Formula (4):(4)$$\theta_{m}=\frac{P}{F}$$
where *P* is the shearing force, N;

*F* is the cutting surface area, m^2^.

### 2.11. Determining the Water-Binding Capacity of Meat

Water-binding capacity (WBC) in meat samples was assessed using a filter paper and planimeter method with a precise pressure-based technique. A 0.3 g ground meat sample was placed on a pre-conditioned filter paper, covered with a polyethylene disk and glass plate, and subjected to a 1 kg weight for 10 min to expel moisture. After pressure application, the resulting spots on the filter paper were measured using a planimeter to determine the areas of the meat and moisture regions. The water-binding capacity was calculated by converting the moisture spot area to water mass (using a conversion factor of 8.4 mg/cm^2^) and expressing the result as a percentage of water retained by the sample. This method provides a quantitative assessment of the meat sample’s ability to retain water under controlled mechanical pressure [29].

### 2.12. Determination of Water-Holding Capacity

To determine the water-holding capacity (WHC), a sample of minced meat weighing 4–6 g was evenly spread with a glass rod on the inner surface of the wide part of the gyrometer. The gyrometer was tightly sealed with a cap and placed in a water bath with the narrow side down for 15 min at the boiling point of water. After that, the mass of the released moisture was determined by the number of readings on the scale of the gyrometer [30].

The water-holding capacity of the meat (WHC, %) was calculated according to Formula (5):*WHC* = *W* − *WRC*,(5)

The water-release capacity (WRC, %) was calculated according to Formula (6):*WRC* = *a n m*^−1^ 100,(6)
where *W* is the total mass fraction of moisture in the sample, %;

*a* is the gyrometer graduation rate, *a* = 0.01 cm^3^;

*n* is the number of graduations;

*m* is the mass of the sample, g.

### 2.13. Determination of Mineral Content

Meat samples were analyzed for mineral content using an ICP-OES atomic emission spectrometer (Spectro, Boschstr, Burghausen, Germany). Meat samples were homogenized thoroughly in a stainless-steel blender to obtain uniform consistency. Approximately 0.5 g of each homogenized sample was weighed accurately into digestion vessels, to which 2 mL of concentrated nitric acid (HNO_3_) was added. Samples were allowed to predigest at room temperature for 2–3 h, after which 1 mL of hydrogen peroxide (H_2_O_2_) was added to facilitate complete oxidation of organic matter. Microwave-assisted digestion was then carried out, gradually increasing the temperature to 200 °C over 40 min, and maintaining at 200 °C for an additional 30 min. After cooling to room temperature, digested samples were transferred to 50 mL volumetric flasks and diluted to volume with ultrapure water. The diluted samples were filtered through ashless filter paper (Whatman Grade 41) to remove any residual particulate matter. The resulting clear solutions were subsequently introduced into the calibrated ICP-OES instrument for mineral quantification. Calibration was performed using appropriate standards to ensure the accuracy and reproducibility of the measurements [31].

### 2.14. Determination of Amino Acid Composition

Amino acid composition was determined using high-performance liquid chromatography (HPLC) with a SHIMADZU LC-20 Prominence system (Shimadzu Corporation, Kyoto, Japan). Meat samples underwent acid hydrolysis with 6 M HCl at 110 °C for 24 h to break down proteins into free amino acids, followed by derivatization with phenylisothiocyanate (PITC) to enhance detection sensitivity. The HPLC analysis utilized a SUPELCO C18 column maintained at 40 °C, with a mobile phase gradient consisting of sodium methanesulfonate buffers and isopropanol-acetonitrile solution. Amino acids were detected at 246 nm and 260 nm using spectrophotometric detection, with identification and quantification performed by comparing sample peak retention times and areas to calibrated amino acid standards from Sigma-Aldrich (St. Louis, MO, USA) [32].

### 2.15. Determination of the Acid Value of Meat Products

The acid value of meat products was determined according to GOST R 55480-2013 using a titrimetric method [33]. A 5–10 g sample of ground meat product was mixed with 50 mL of a neutralized spirit-ether solvent (1:1 ethanol-diethyl ether mixture) and thoroughly agitated for 5 min. The mixture was heated in a water bath at 50–60 °C for 10 min to extract free fatty acids, then cooled. The extract was titrated with 0.1 M sodium hydroxide solution using phenolphthalein as an indicator, with titration continuing until a stable pink color appeared. The acid value (mg KOH/g) was calculated using Formula (7):(7)$$X=\frac{V\cdot K\cdot 5.61}{m}$$*V*—volume of 0.1 mol/dm^3^ potassium hydroxide solution consumed for titration, cm^3^;*K*—correction factor to the nominal concentration of solutions;*m*—mass of fat in the analyzed sample, g;5.61—mass of potassium hydroxide corresponding to 1 cm^3^ 0.1 mol/dm^3^ of potassium hydroxide solution, mg.

### 2.16. Texture Profile Analysis

Texture Profile Analysis (TPA) was performed using a Brookfield CT3 Texture Analyzer (AMETEK) equipped with a control unit, measuring head, appropriate fixtures, and a portable vertical stage (TA-RT-KIT). Prior to testing, samples were cut into uniform cubes (10 mm × 10 mm × 10 mm), placed in aluminum cylinders, and equilibrated to a temperature of 20–25 °C. The pinstripe was configured with a compression speed of 5 mm/s and set to compress each sample to 75% of its initial height. Each sample underwent two consecutive compression cycles. From the recorded force–deformation curves, hardness was determined as the maximum peak force during the first compression. Cohesiveness was calculated as the ratio of the area under the second compression curve (A2) to that under the first compression curve (A1), reflecting the sample’s structural integrity after the initial deformation. Springiness, defined as the extent to which the sample recovered its height after deformation, was measured by comparing the distances the probe traveled in each cycle. Adhesiveness was calculated from the negative area of the force–time curve after the first compression, reflecting the work required to overcome the attractive forces between the sample and the probe. Throughout testing, force and deformation data were continuously recorded, ensuring accurate quantification of the textural parameters.

### 2.17. Microstructure

Observations of the microstructure of meat–bone paste were made using a low vacuum scanning electron microscope JSM-6390LV JEOL (Tokyo, Japan). The accelerating voltage applied to the electron beam was 15 KV at a magnification of ×100 [34].

### 2.18. Statistical Analysis

The experiments were performed in triplicate. Standard deviation values were indicated for all measurements. Differences in the measurements of the experimental and control groups were calculated using analysis of variation (one-way ANOVA) using the Tukey test. A *p*-value of ˂0.05 was considered significant. Statistical analysis was carried out using Excel 2016 (Microsoft Corporation, Redmond, Washington, DC, USA) and Statistica 12 PL (StatSoft, Inc., Tulsa, OK, USA) software packages.

## 3. Results and Discussion

### 3.1. Effect on Microbiological Safety

The elimination of Clostridium spp. in beef is paramount for extending shelf life and ensuring product safety. In this study, untreated control samples (Sample 1) showed the presence of *Clostridium* spp., whereas all irradiated samples (3, 6, and 9 kGy) tested negative (Table 3). This finding underscores the efficacy of ionizing radiation in reducing or eliminating pathogenic and spoilage microorganisms. From a practical standpoint, the dose of 3 kGy was sufficient to achieve complete inactivation of Clostridium spp. under the conditions tested, suggesting it may be the minimal effective dose in this context. The higher doses (6 and 9 kGy) also resulted in no detectable *Clostridium* spp., which supports the robustness of higher-dose irradiation but must be weighed against potential effects on meat quality.

The efficacy of irradiation depends on the resistance of the target organisms. Vegetative bacteria and yeasts are relatively easily destroyed, whereas bacterial spores and viruses require higher doses. Spore-forming bacteria (like *Clostridium* or *Bacillus* species) are quite radiation-resistant; however, even spores are significantly inactivated by sufficiently high doses (on the order of 5–10 kGy or more) [35]. *Clostridium* bacteria (especially their spores) are among the most radiation-resistant food pathogens, but high-dose irradiation can inactivate them. The study of Parry-Hanson et al. [36] found that a 9 kGy gamma dose completely eliminated *Clostridium perfringens* spores inoculated on ready-to-eat beef tripe.

### 3.2. Total Viable Count

The effect of electron-beam irradiation on the total number of mesophilic aerobic and facultatively anaerobic microorganisms (Total Viable Count, TVC) was evaluated. TVC, measured in colony-forming units per gram (CFU/g), is a key microbiological indicator widely used to assess microbial quality, safety, and shelf-life potential of meat products. The irradiation doses applied (3, 6, and 9 kGy) effectively reduced microbial contamination in beef samples. The lowest dose (3 kGy) substantially decreased the initial microbial load (from 300 to 100 CFU/g, a 67% reduction) (Figure 3). However, increasing the irradiation dose from 3 kGy to 6 kGy did not result in a further measurable reduction in the microbial count. This plateau suggests that the microorganisms remaining after a 3 kGy dose likely include radiation-resistant strains, spores, or populations shielded by the meat matrix or fat content. Complete microbial inactivation was achieved at a 9 kGy dose, suggesting this dose is sufficiently robust to eliminate mesophilic aerobic and facultatively anaerobic organisms, including potential spoilage bacteria and pathogens commonly found in meat (Figure 4).

Ionizing radiation kills microorganisms by damaging their genetic material. High-energy photons or electrons break chemical bonds in DNA (direct effect) and generate reactive species from water that further attack cellular components (indirect effect) [37].

In practical terms, even a low dose (~1–3 kGy) can achieve several log_10_ reductions in common meat-borne bacteria. For example, doses in the range of 2–3 kGy are sufficient to significantly reduce or eliminate pathogens like *Escherichia coli* O157:H7, *Salmonella*, *Campylobacter*, *Staphylococcus aureus*, and *Listeria monocytogenes* in fresh meat [35].

A study on chilled poultry reported that 2.5 kGy and 5.0 kGy treatments reduced surface microflora by about 3–4 log cycles, which translated into extending the shelf life to ~2–3 weeks under refrigeration [38]. Similarly, researchers in Uruguay found that using up to 2.5 kGy on beef trimmings gave ~2 log CFU/g reduction in *L. monocytogenes* and ~5 log reduction in *E. coli* O157:H7, while 5 kGy achieved even greater pathogen lethality [39].

Gamma irradiation at moderate doses (~3 kGy) has been shown to suppress or kill common pathogens like *Salmonella*, *E. coli*, *Listeria monocytogenes*, and spoilage microbes on beef [40]. Sedeh et al. observed that a 3 kGy dose in bovine meat effectively reduced mesophilic bacteria, coliforms, and *Staphylococcus aureus*, improving microbial safety [41]. This aligns with extensive research showing that ionizing radiation can significantly reduce or eliminate pathogenic bacteria and spores in meat [42].

### 3.3. Effect on Acid Number

The acid number, an indicator of lipid hydrolysis and potential rancidity, exhibited an upward trend with higher irradiation doses. The control sample showed an acid number of 0.42, whereas samples irradiated with 3, 6, and 9 kGy had values of 0.45, 0.60, and 0.65, respectively (Figure 5). Electron-beam irradiation can induce oxidative and hydrolytic changes in lipids, thereby increasing free fatty acid content. When high-energy rays pass through meat, they ionize water molecules and generate reactive oxygen species. These radicals can attack fatty acids, initiating lipid peroxidation chains. One immediate effect is the cleavage of triglycerides into glycerol and free fatty acids, raising the acid value [35]. These oxidative changes can also lead to off-flavors if not controlled or mitigated with proper packaging and storage conditions.

Despite the higher acid number in samples irradiated at 9 kGy, the absolute values remained within a range that may be acceptable for meat products, though this threshold depends on product type, target market regulations, and desired shelf-life characteristics.

### 3.4. Effect on Protein and Amino Acids Content

The protein content of the control sample was 25.06 g/100 g, which decreased progressively with increasing irradiation dose. Samples irradiated with 3 kGy, 6 kGy, and 9 kGy contained 22.33, 20.91, and 18.95 g/100 g of protein, respectively (Table 4). This downward trend can be attributed to several irradiation-induced reactions, including protein denaturation or fragmentation. High-energy electrons can cause breaks in polypeptide chains and lead to oxidative modifications of amino acids. Another consequence of irradiation on proteins is reduced extractability and solubility of certain protein fractions. Early research on irradiated chicken meat found that myosin (a major muscle protein) became less extractable after ~5 kGy, suggesting some protein denaturation or aggregation [38].

While a modest decrease in protein content may be tolerable from a nutritional perspective, a more pronounced decline at higher doses (i.e., 9 kGy) becomes a concern for both nutritional value and functional properties. Thus, balancing microbial safety with minimal nutrient deterioration is a crucial consideration in irradiated meat processing.

The data reveal a dose-dependent effect on both individual amino acids and the total amino acid content. In the control sample, the total amino acid content was 22.34 g/100 g, which progressively declined to 21.97 g/100 g at 3 kGy, 19.64 g/100 g at 6 kGy, and 16.87 g/100 g at 9 kGy (Table 4). This trend suggests that higher irradiation doses lead to significant degradation of amino acids, a finding that is consistent with the literature on protein irradiation.

Individual amino acids responded differently to irradiation. For example, arginine decreased sharply from 4.71 g/100 g in the control to 3.75 g/100 g at 3 kGy and further to 2.11 g/100 g at 6 kGy, with a slight rebound at 9 kGy (2.66 g/100 g). In contrast, lysine showed a modest increase at 3 kGy (2.63 g/100 g) and 6 kGy (2.66 g/100 g) relative to the control (2.43 g/100 g) but then declined at 9 kGy (2.24 g/100 g). Similar non-linear trends were observed with other amino acids, such as phenylalanine, histidine, and the branched-chain amino acids (leucine + isoleucine and valine), where moderate doses sometimes led to slight increases or minor fluctuations before a more pronounced decrease at the highest dose. The increase in some amino acids at 3 and/or 6 kGy may be explained by protein unfolding. Ionizing radiation induces partial denaturation, exposing previously buried amino acid residues, which enhances their detection during analysis. This likely reflects improved analytical recovery rather than a net increase in amino acid concentration, a phenomenon consistent with protein unfolding observed in irradiated meat. In study [43], electron-beam irradiation of meat at 7 kGy did not significantly change total amino acid content, whereas a higher dose of 9 kGy caused a measurable decrease in total amino acids.

These observations indicate that moderate irradiation can cause partial protein breakdown, potentially releasing free amino acids and transiently increasing their measurable concentrations. However, as the irradiation dose increases, the accumulation of free radicals and oxidative reactions predominates, leading to the degradation and modification of amino acids. For instance, sulfur-containing amino acids (e.g., methionine) and aromatic amino acids (e.g., tyrosine and phenylalanine) are particularly susceptible to oxidative damage, which explains their decline at higher doses [44]. Free radical attacks can oxidize side chains or break peptide bonds, converting amino acids like lysine, arginine, proline, cysteine, threonine, leucine, and histidine into carbonyl derivatives or other altered compounds [45]. The overall reduction in total amino acid content suggests that excessive irradiation may compromise the nutritional quality of meat by degrading essential amino acids.

The progressive reduction in total amino acid content—from 22.34 g/100 g in the control to 16.87 g/100 g at 9 kGy—raises concerns regarding the nutritional quality of irradiated meat. Essential amino acids are crucial for human health, and their degradation may diminish the meat’s overall protein quality. Balancing microbial safety with nutritional retention is therefore paramount. While lower irradiation doses may provide sufficient pathogen control with minimal nutritional loss, high doses, although more effective in microbial inactivation, could compromise the meat’s nutritional value.

Radiation treatment of meat primarily affects sulfur-containing and aromatic amino acids through oxidation and deamination. Protein cross-linking occurs from free radical formation, altering texture. While higher doses progressively degrade amino acids, including essential ones, approved irradiation levels typically cause minimal nutritional impact. These changes are dose-dependent and influence meat’s sensory qualities.

### 3.5. Effect on Mineral Content

In the control sample, the baseline iron content was 2.83 mg/100 g, and zinc was 3.75 mg/100 g. Following irradiation, a dose-dependent response was observed. At 3 kGy, iron content increased slightly to 2.95 mg/100 g, while zinc experienced a modest decrease to 3.55 mg/100 g. With a further increase in dose to 6 kGy, a more pronounced rise was noted for both minerals: iron increased to 3.92 mg/100 g and zinc to 3.97 mg/100 g. At the highest dose of 9 kGy, the iron level declined to 2.8 mg/100 g, effectively reverting to the control level, whereas zinc reached 4.01 mg/100 g (Figure 6).

These trends indicate that irradiation affects the mineral composition in a complex and element-specific manner. The observed increase in iron at moderate doses, followed by a decline at the highest dose, contrasts with the behavior of zinc, which exhibits a slight initial reduction at 3 kGy but then progressively increases at higher doses.

The observed changes in mineral content likely stem from irradiation-induced modifications in the meat matrix. Ionizing radiation is known to generate free radicals that can disrupt cellular structures and denature proteins. Such alterations can affect the binding and release of minerals from protein complexes and intracellular stores. For iron, the moderate increase at 3 kGy and more pronounced elevation at 6 kGy may be attributed to the disruption of myoglobin and other iron-containing proteins, resulting in a transient increase in extractable or measurable iron. However, at 9 kGy, the decrease to control levels might suggest further degradation or the formation of insoluble iron complexes that are less amenable to standard analytical methods.

In contrast, zinc’s behavior demonstrates a different sensitivity to irradiation. The initial decrease at 3 kGy suggests that early irradiation may induce conformational changes in zinc-binding sites, transiently reducing its extractability. As the irradiation dose increases to 6 and 9 kGy, the subsequent breakdown of these altered complexes may release zinc, leading to recovery and even a slight enhancement in its measurable content relative to the control.

### 3.6. Effect on Water-Holding Capacity

Water-holding capacity (WHC) is a critical parameter influencing the juiciness and overall palatability of meat. The control sample demonstrated a WHC of 65.8%, which decreased to 64.2% (3 kGy), 61.3% (6 kGy), and 57.8% (9 kGy). These results suggest that higher irradiation doses can impair the structural integrity of myofibrillar proteins, leading to reduced water retention in muscle tissue (Figure 7). Even though the drop from control to 3 kGy was modest, the difference became more pronounced at higher doses.

From a processing perspective, diminished WHC can affect product yield and consumer acceptability. Meat processors typically aim to balance the benefits of microbial safety with maintaining quality traits such as juiciness.

### 3.7. Textural Changes in Beef Due to Irradiation

Shear force measurements serve as an indicator of meat tenderness. The control sample had a shear force of 157.3 Pa, which fell to 154.1 Pa (3 kGy), 146.3 Pa (6 kGy), and 132.13 Pa (9 kGy). These data indicate that electron-beam irradiation generally decreased shear force, implying an increase in tenderness (Table 5). The structural modifications in myofibrillar proteins caused by irradiation may account for this tenderization effect. However, at higher doses (e.g., 9 kGy), excessive protein denaturation might compromise other quality attributes, such as flavor and WHC. A study on pork found that increasing gamma doses (3–7 kGy) caused greater degradation of myofibrillar proteins and collagen, resulting in a lower Warner–Bratzler shear force (i.e., improved tenderness) immediately after treatment [46].

The influence of irradiation was clearly dose-dependent, indicating significant structural changes in meat proteins that impacted textural properties.

Hardness measures the force required to compress meat samples, directly relating to sensory perceptions of tenderness. In the present study, hardness decreased progressively with increasing irradiation dose, from 58.54 N in the untreated control to 53.87 N (at 6 kGy) and finally 50.11 N at the highest irradiation dose (9 kGy). Such a consistent reduction in hardness is likely attributable to the partial breakdown of structural proteins, particularly myofibrillar proteins and collagen, due to the radiolytic cleavage of peptide bonds, which can weaken muscle fibers and connective tissues.

Springiness represents the meat’s ability to return to its original shape after deformation, while cohesiveness reflects how well the meat structure withstands mechanical deformation. These attributes strongly influence meat’s mouthfeel. The data obtained show that springiness in the control (non-irradiated) sample was higher (0.86–0.91 typically reported in literature), whereas post-irradiation samples (at 3 and 6 kGy) stabilized at 0.80 mm, signifying slight reductions. The notable drop in cohesiveness at the highest dose (9 kGy, from 0.88 at 3 kGy downwards) further indicates structural weakening due to irradiation-induced protein and connective tissue alterations.

Reduced springiness and cohesiveness at higher doses typically reflect protein degradation, including partial breakdown of collagen and denaturation of muscle fibers.

Gumminess (the energy required to disintegrate a semi-solid meat structure) and chewiness (energy required to masticate the meat before swallowing) showed a notable downward trend following irradiation. Gumminess decreased from 48.34 N at 3 kGy to 34.12 N at 9 kGy, and chewiness (a combined effect of hardness, cohesiveness, and springiness) decreased markedly from 38.51 N·mm (3 kGy) to 23.89 N·mm (9 kGy), confirming a significant softening effect.

Moderate irradiation (3–6 kGy) may positively influence tenderness with minimal adverse sensory impact. However, higher doses (9 kGy or more) lead to substantial structural damage, potentially reducing consumer acceptance due to compromised texture and perceived freshness.

### 3.8. Microstructural Analysis of Beef Before and After Irradiation Treatment

The initial structural morphology of beef muscle tissue (control sample, untreated) was examined by electron microscopy (Figure 8). The sample displayed dense bundles composed predominantly of tightly interconnected polygonal muscle fibers, forming a well-organized and compact tissue structure. Between these fiber bundles, narrow connective tissue interfaces appeared as dark inter bundle spaces, varying slightly in width. Furthermore, fibrous filaments resembling a web-like structure were observed, overlaying and partially enclosing the muscle fiber bundles.

The micrograph reveals intact tissue organization, indicating structural integrity before irradiation treatment. The observed densely packed fibers and intact connective tissue connections correlate typically with non-processed meat samples as described in prior research literature. The clearly discernible tight adhesion and structural compactness of fiber bundles are indicative of intact myofibrillar proteins and connective tissues, reflecting the absence of structural disruption prior to irradiation.

These morphological characteristics, notably the compact arrangement and integrity of muscle fibers and connective tissue, are commonly associated with favorable sensory and functional attributes, such as higher water-holding capacity, textural firmness, and resistance to external mechanical forces.

The described structural morphology thus serves as a baseline reference for subsequent evaluations of irradiation-induced changes in meat tissue, facilitating accurate assessment of structural alterations due to electron-beam irradiation.

Post-irradiation analysis revealed significant structural changes compared to the untreated control. Specifically, the muscle fiber bundles were observed to be less densely packed, showing a notable reduction in their compactness. While polygonal muscle fibers within each bundle remained tightly connected, the overall bundle integrity appeared compromised, characterized by less densely arranged fiber clusters and increased spacing between bundles.

Distinct connective tissue interfaces, visible as dark bands of variable widths between fiber bundles, were clearly identified. Moreover, fibrous filaments resembling web-like threads were present, though their arrangement appeared less uniform compared to the untreated samples. Notably, irradiation-induced modifications led to looser adherence and partial disruption of the inter-bundle connective tissue network, thus diminishing the tightness of the overall muscular structure.

These microstructural changes highlight the impact of irradiation on muscle tissue organization. The observed loosening and formation of interstitial spaces likely result from partial denaturation or fragmentation of structural proteins (e.g., collagen, myofibrillar proteins) induced by radiolytic reactions and associated oxidative mechanisms triggered by ionizing radiation.

The microstructural alterations documented here align closely with existing literature, reinforcing established correlations between irradiation-induced structural modifications and altered textural characteristics, such as reduced hardness, decreased water-holding capacity, and increased tenderness. These microstructural findings provide a clear scientific explanation for the changes previously noted in textural and physicochemical parameters of irradiated meat. Irradiation can induce ultrastructural changes in muscle tissue that influence meat texture. Microscopic analyses have observed shrinkage and disruption in muscle fibers after irradiation. For example, Yoon (2003) reported that irradiating chicken breast (~2–3 kGy) led to a significant reduction in sarcomere width (myofibril unit size) and noticeable fragmentation of myofibrils [47].

### 3.9. Economic Feasibility, Consumer Acceptance, and Regulatory Implications

Economic feasibility is promising since irradiation reduces spoilage, extends shelf life, and lowers recall costs, which can offset the initial capital expenditure for modern irradiation facilities. Consumer acceptance, while historically cautious due to misconceptions regarding radioactivity and quality alterations, is expected to improve through targeted education and clear labeling—factors that further drive market competitiveness.

In the Republic of Kazakhstan, irradiated meat is regulated by the Eurasian Economic Union (EAEU) technical regulations [48,49] and as well as by the national standard [50], which defines identification methods for irradiation. Sanitary control is performed in compliance with the country’s regulations on radiation safety. Consequently, when these established normative requirements are met, the production and consumption of irradiated meat are considered acceptable and safe for consumers, as confirmed by the relevant regulatory documents. This robust framework, combined with economic and consumer factors, underscores the viability of adopting irradiation technology in Kazakhstan’s meat industry.

## 4. Conclusions

This study demonstrates that electron-beam irradiation is an effective strategy for enhancing the safety of beef, with a 3 kGy dose being sufficient to inhibit the development of pathogens. At this moderate dose, the total viable counts were significantly reduced, ensuring improved microbial safety without the detrimental impacts associated with higher doses. Importantly, processing at 3 kGy maintained the integrity of the protein profile and preserved the amino acid composition, indicating that essential nutritional qualities remain intact. Additionally, samples irradiated at 3 kGy exhibited lower levels of oxidation, minimizing the risk of oxidative damage that can negatively affect flavor, color, and overall quality. These findings suggest that a 3 kGy dose optimally balances microbial inactivation with the preservation of meat’s nutritional and sensory attributes. Consequently, implementing a 3 kGy irradiation protocol can serve as a viable approach in commercial meat processing, providing effective pathogen control while avoiding significant deterioration of protein structure and amino acid content. Future research should focus on further validating these results across different meat types and processing conditions to reinforce the use of 3 kGy as a standard treatment for ensuring both food safety and quality preservation.

## Figures and Tables

**Figure 1 foods-14-01460-f001:**
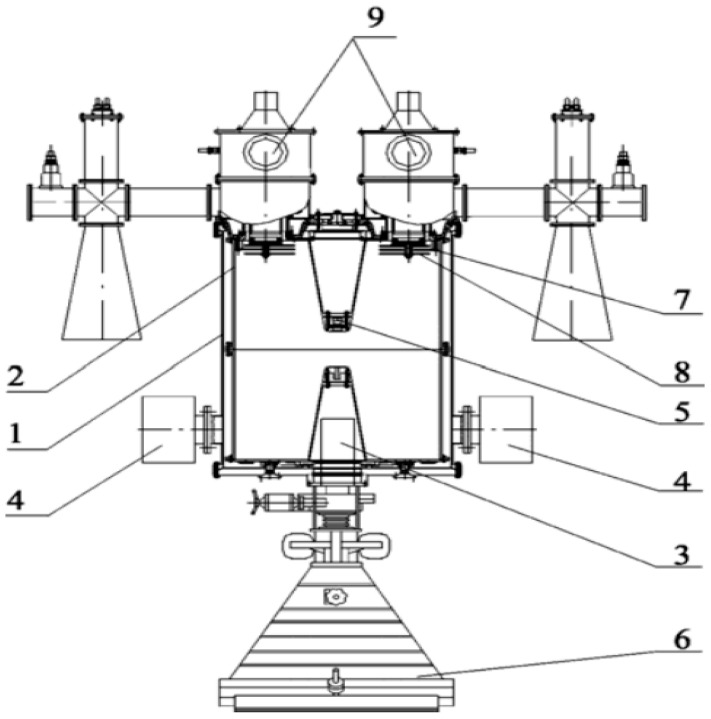
The pulsed linear electron accelerator ILU-10. 1—vacuum tank; 2—resonator; 3—focusing lens; 4—magnetic discharge pumps; 5—electron injector; 6—discharge device; 7—coupling loop; 8—vacuum capacitor of the coupling loop; 9—RF generator.

**Figure 2 foods-14-01460-f002:**
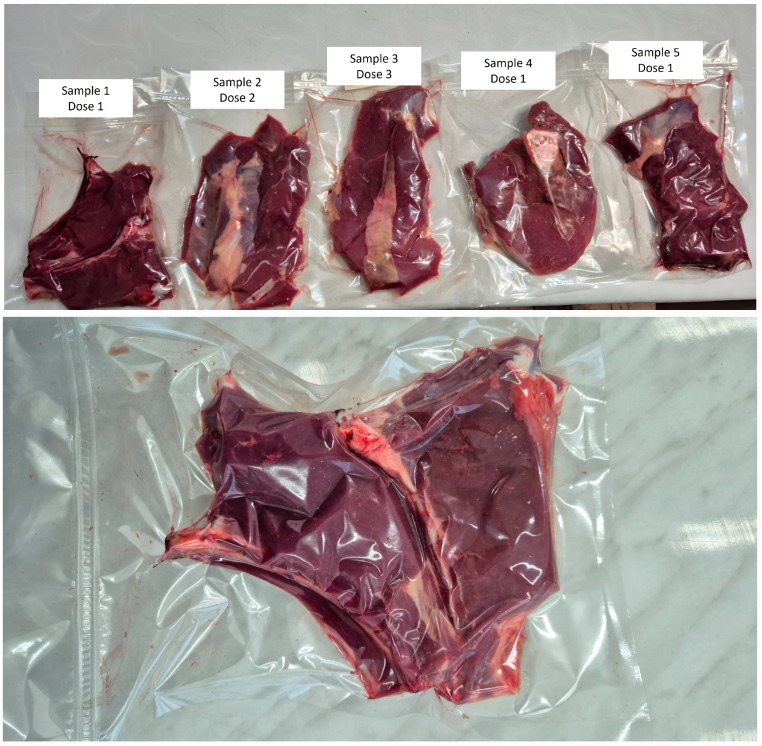
Vacuum-packed meat samples.

**Figure 3 foods-14-01460-f003:**
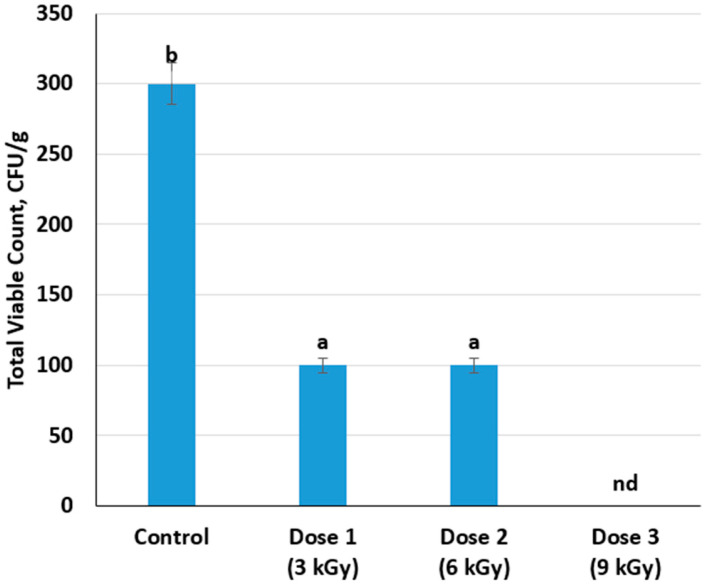
Variation of the total viable count depending on the dose of radiation treatment of meat (Different lowercase letters (a,b) indicate statistically significant differences between the samples (*p* < 0.05)). nd: not detected.

**Figure 4 foods-14-01460-f004:**
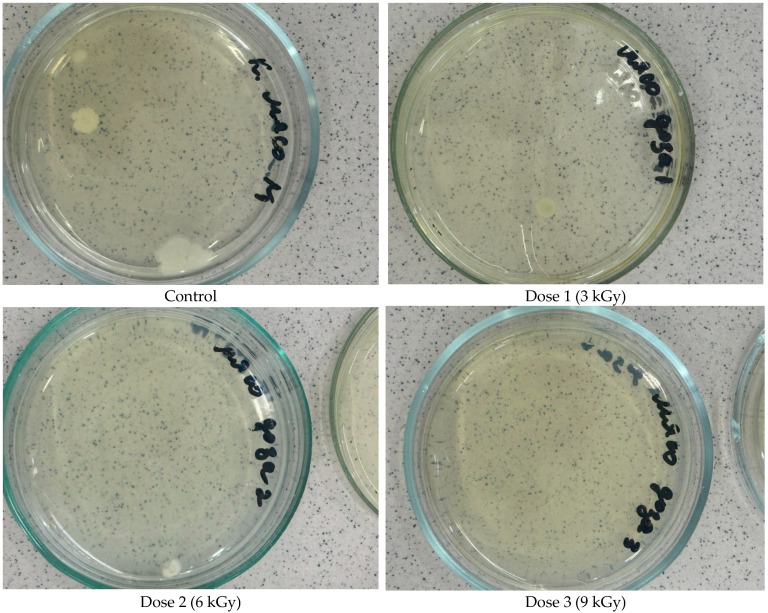
Growth of bacterial colonies on agar medium in a Petri dish depending on the dose of radiation treatment of meat.

**Figure 5 foods-14-01460-f005:**
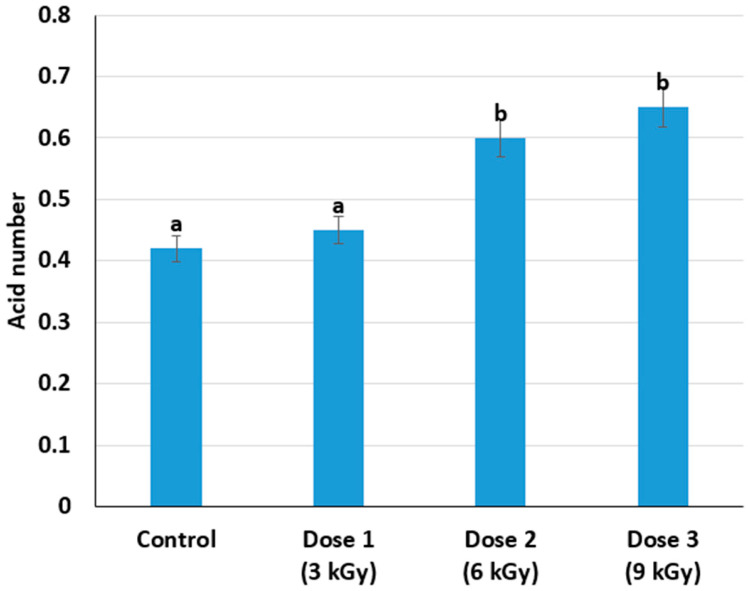
Change in acid number depending on the dose of radiation treatment of meat (different lowercase letters (a,b) indicate statistically significant differences between the samples (*p* < 0.05)).

**Figure 6 foods-14-01460-f006:**
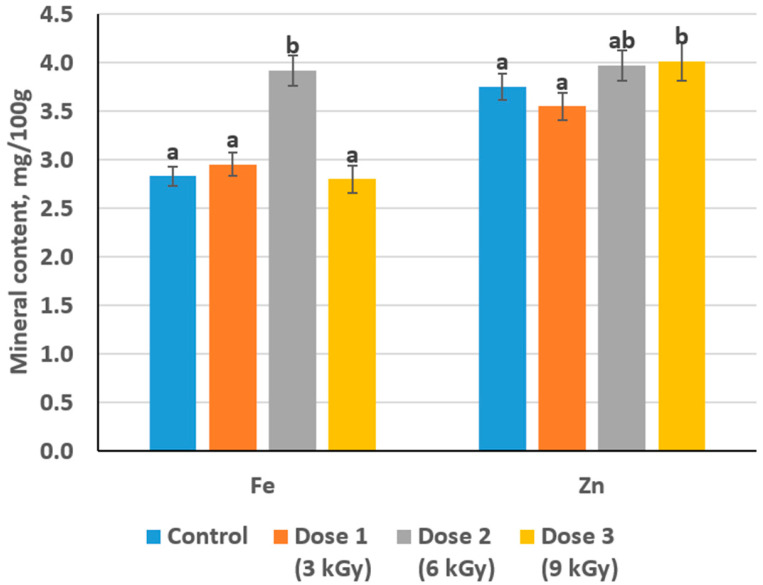
The mineral content of meat samples depending on the dose of radiation treatment (different lowercase letters (a,b) indicate statistically significant differences between the samples (*p* < 0.05)).

**Figure 7 foods-14-01460-f007:**
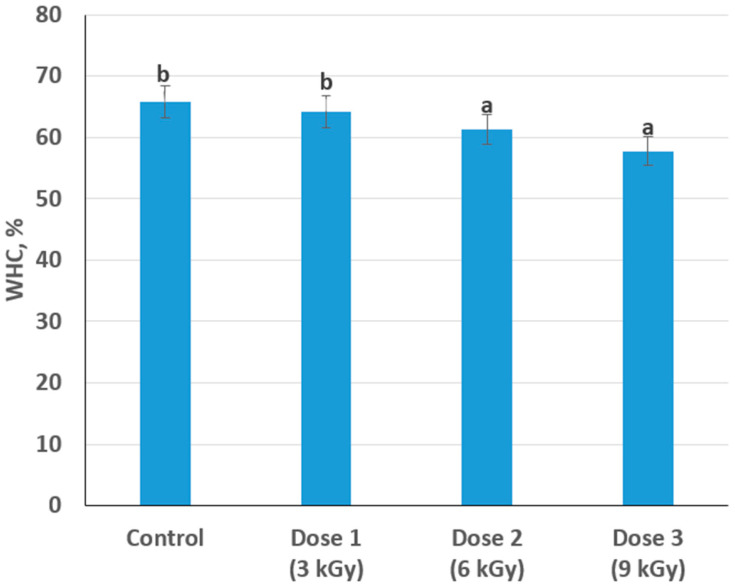
Change in water-holding capacity depending on the dose of radiation treatment of meat (different lowercase letters (a,b) indicate statistically significant differences between the samples (*p* < 0.05)).

**Figure 8 foods-14-01460-f008:**
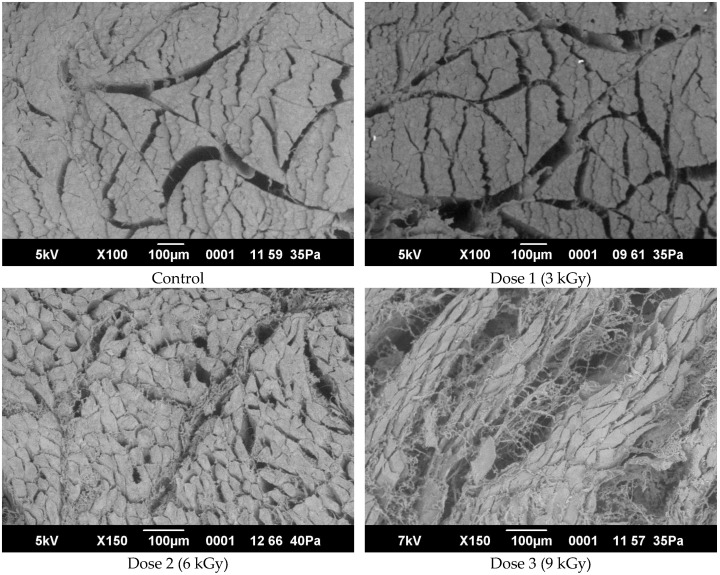
Microstructural analysis of beef before and after irradiation treatment.

**Table 1 foods-14-01460-t001:** Irradiation parameters for ILU-10 accelerator.

Parameters	ILU-10
Energy range, MeV	2.5–5.0
Electron-beam power (max), kW	50
Maximum beam current (max), mA	20
Processing depth (electrons)	4 g/cm^2^
Productivity at 1 kGy dose (electrons)	90,000 kg/h
Processing depth (bremsstrahlung, 1 side)	30 g/cm^2^
Productivity at 1 kGy dose (bremsstrahlung)	4500 kg/h

**Table 2 foods-14-01460-t002:** Main technological process parameters.

Parameter	Value
Electron energy, MeV	2.5–5
Electron-beam power (max), kW	50
Average beam current, mA	up to 10
Current adjustment limits, mA	0–10
Energy instability	±2%
Beam current instability	±2%
Scan width, mm	800
Power extracted beam (unevenness ≤ ±10% over 800 mm)	≥80% accelerated beam power
Transport line speed, cm/s	2–8
Conveyor speed deviation from set value	≤3%

**Table 3 foods-14-01460-t003:** Effect of meat processing on the presence of *Clostridium* spp.

Name of the Microorganism	Before Treatment	Dose 1 (3 kGy)	Dose 2 (6 kGy)	Dose 3 (9 kGy)
*Clostridium* spp. 0.01 g	Detected	Not detected	Not detected	Not detected

**Table 4 foods-14-01460-t004:** Protein and amino acid content of non- and irradiated meat, g/100 g (mean ± SD).

Indicator	Control	Dose 1 (3 kGy)	Dose 2 (6 kGy)	Dose 3 (9 kGy)
Protein	25.06 ± 0.46 ^d^	22.33 ± 0.35 ^c^	20.91 ± 0.43 ^b^	18.95 ± 0.38 ^a^
Amino acids				
Arginine	4.71 ± 0.08 ^d^	3.75 ± 0.06 ^c^	2.11 ± 0.03 ^b^	2.66 ± 0.04 ^a^
Lysine	2.43 ± 0.03 ^b^	2.63 ± 0.05 ^c^	2.66 ± 0.05 ^c^	2.24 ± 0.02 ^a^
Tyrosine	1.03 ± 0.01 ^b^	1.08 ± 0.02 ^c^	1.11 ± 0.02 ^c^	0.77 ± 0.01 ^a^
Phenylalanine	1.31 ± 0.02 ^c^	1.38 ± 0.02 ^d^	1.22 ± 0.02 ^b^	1.06 ± 0.01 ^a^
Histidine	0.80 ± 0.01 ^b^	0.98 ± 0.02 ^c^	0.77 ± 0.01 ^b^	0.67 ± 0.01 ^a^
Leucine+isoleucine	1.71 ± 0.03 ^b^	1.88 ± 0.03 ^c^	1.66 ± 0.02 ^b^	1.40 ± 0.02 ^a^
Methionine	0.97 ± 0.02 ^c^	0.91 ± 0.01 ^b^	0.95 ± 0.02 ^c^	0.74 ± 0.01 ^a^
Valine	1.71 ± 0.02 ^bc^	1.75 ± 0.04 ^c^	1.66 ± 0.02 ^b^	1.40 ± 0.02 ^a^
Proline	1.57 ± 0.02 ^c^	1.50 ± 0.03 ^c^	1.44 ± 0.02 ^b^	1.13 ± 0.01 ^a^
Threonine	1.57 ± 0.03 ^c^	1.38 ± 0.02 ^b^	1.55 ± 0.02 ^c^	1.26 ± 0.02 ^a^
Serine	1.09 ± 0.01 ^b^	1.25 ± 0.02 ^c^	1.05 ± 0.02 ^b^	0.91 ± 0.01 ^a^
Alanine	1.86 ± 0.02 ^b^	2.00 ± 0.04 ^c^	2.00 ± 0.04 ^c^	1.54 ± 0.02 ^a^
Glycine	1.57 ± 0.03 ^c^	1.50 ± 0.02 ^c^	1.44 ± 0.02 ^b^	1.08 ± 0.02 ^a^
Total	22.34	21.97	19.64	16.87

^a–d^ Means within the same row with different letters differing significantly among samples (*p* < 0.05). Values are expressed as the mean ± SD.

**Table 5 foods-14-01460-t005:** Texture analysis of non- and irradiated meat.

Parameter	Control	Dose 1 (3 kGy)	Dose 2 (6 kGy)	Dose 3 (9 kGy)
Hardness (N)	58.54 ± 0.73 ^c^	55.14 ± 0.62 ^bc^	53.87 ± 0.55 ^b^	50.18 ± 1.03 ^a^
Springiness (mm)	0.86 ± 0.02 ^c^	0.80 ± 0.01 ^b^	0.80 ± 0.01 ^b^	0.70 ± 0.01 ^a^
Cohesiveness	0.91 ± 0.01 ^d^	0.88 ± 0.02 ^c^	0.75 ± 0.01 ^b^	0.68 ± 0.01 ^a^
Gumminess (N)	53.46 ± 0.78 ^d^	48.34 ± 0.67 ^c^	40.40 ± 0.42 ^b^	34.12 ± 0.56 ^a^
Chewiness (N × mm)	45.80 ± 0.72 ^d^	38.51 ± 0.38 ^c^	32.32 ± 0.45 ^b^	23.89 ± 0.44 ^a^
Shear force (N)	157.3 ± 2.13 ^c^	154.1 ± 2.04 ^c^	146.3 ± 3.42 ^b^	132.13 ± 1.80 ^a^

^a–d^ Means within the same row with different letters differing significantly among samples (*p* < 0.05). Values are expressed as the mean ± SD.

## Data Availability

The original contributions presented in the study are included in the article. Further inquiries can be directed to the corresponding author.

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
