# Peer review of "Effect of Electron-Beam Irradiation on Microbiological Safety, Nutritional Quality, and Structural Characteristics of Meat"

_foods, 2025, doi:10.3390/foods14091460_

Round 1
Reviewer 1 Report
Comments and Suggestions for Authors
C1.
Line 43 – 45
Dear authors, please provide reference for the following statement „Against this backdrop, food irradiation has emerged as a promising non-thermal technology that enhances food safety, extends shelf life, and minimizes nutritional losses.“
C2.
Line 80 – 82
Please change sentence “In the work [19], minced poultry meat irradiated at a dose of 30 kGy (using 60Co) exhibited significant deterioration in organoleptic properties, characterized by a gray color, a burning odor, and an overall decline in visual quality.” to “In the work of Donskova and Belyaev [19], minced poultry meat irradiated at a dose of 30 kGy (using 60Co) exhibited significant deterioration in organoleptic properties, characterized by a gray color, a burning odor, and an overall decline in visual quality.”
C3.
Line 88-90
I kindly ask the authors to provide reference for the following statement “Studies have shown that public understanding significantly impacts acceptance; providing scientifically accurate information and clear labeling (e.g., the Radura symbol) can significantly improve consumer perception and acceptance.”
C4.
Line 129
Please provide the following information in section 2.3.
- Which meat cuts were used in study?
- What was the average weight of each sample packed in vacuum bags?
- How long after cooling the experiment was conducted?
C5.
Line 288 – 290
Please explain in more detail the process of decomposition in a microwave. Have you used nitric acid, what temperature stages were applied during these 12 hours? In my opinion, 12 hours seems like too much time for decomposition, this usually lasts up to one hours. However, in any case, please provide reference for this method and provide information regarding microwave producers.
C6.
Line 364
In section 2.18. (Statistical Analysis) please provide information which software was used for statistical analysis of obtained data in the study.
C7.
Line 373 – 374
The sentence “The study [35] found that a 9 kGy gamma dose completely eliminated Clostridium perfringens spores inoculated on ready-to-eat beef tripe.” should be changed to “The study of Parry-Hanson et al. [35], found that a 9 kGy gamma dose completely eliminated Clostridium perfringens spores inoculated on ready-to-eat beef tripe.”
C8.
Line 413
I kindly ask the authors to use Italics for bacteria species. In sentence at line 413, Staphylococcus aureus should have been written in Italics, Staphylococcus aureus. Please check the entirety of the manuscript for similar mistakes.
C9.
Line 451
In table 4. Please explain the meaning of ±0.46. I believe that this represents standard deviation. However, this should be explained both in table and caption.
C10.
Line 576
Similar to in table 4., in table 5, please explain the meaning of ± symbol and provide explanation in both, table and caption.
Author Response
Reviewer 1
|
Comment |
Answer |
|
C1. Line 43 – 45 Dear authors, please provide reference for the following statement „Against this backdrop, food irradiation has emerged as a promising non-thermal technology that enhances food safety, extends shelf life, and minimizes nutritional losses.“ |
Added reference:
|
|
C2. Line 80 – 82 Please change sentence “In the work [19], minced poultry meat irradiated at a dose of 30 kGy (using 60Co) exhibited significant deterioration in organoleptic properties, characterized by a gray color, a burning odor, and an overall decline in visual quality.” to “In the work of Donskova and Belyaev [19], minced poultry meat irradiated at a dose of 30 kGy (using 60Co) exhibited significant deterioration in organoleptic properties, characterized by a gray color, a burning odor, and an overall decline in visual quality.” |
Line 79 Corrected |
|
C3. Line 88-90 I kindly ask the authors to provide reference for the following statement “Studies have shown that public understanding significantly impacts acceptance; providing scientifically accurate information and clear labeling (e.g., the Radura symbol) can significantly improve consumer perception and acceptance.” |
Added
|
|
C4. Line 129 Please provide the following information in section 2.3. - Which meat cuts were used in study? - What was the average weight of each sample packed in vacuum bags? - How long after cooling the experiment was conducted? |
Line 112-115: The meat samples were obtained immediately after slaughtering of cows (2-2.5 years old) from the peasant farm “Mukinov” of Ernazar village, Beskaragai district, Abai region (Kazakhstan). The meat obtained after slaughter was cooled in refrigerat-ed chambers at 0–(+2) °C for 24 h. After cooling meat was vacuum-packed and frozen to -18 °C.
Line 138-141: Beef meat samples (Longissimus dorsi muscle) weighing approximately 150-200 grams each were individually sealed under vacuum conditions (40–60 mm Hg) in polyethylene bags measuring 20×25 cm (Figure 2). Average chemical composition - protein 25%, fat 15%, moisture 59%. |
|
C5. Line 288 – 290 Please explain in more detail the process of decomposition in a microwave. Have you used nitric acid, what temperature stages were applied during these 12 hours? In my opinion, 12 hours seems like too much time for decomposition, this usually lasts up to one hours. However, in any case, please provide reference for this method and provide information regarding microwave producers.
|
Revised Line 316-330: Meat samples were analyzed for mineral content using an ICP-OES atomic emission spectrometer (Spectro, Boschstr, Burghausen, Germany). Fresh meat samples were homogenized thoroughly in a stainless-steel blender to obtain uniform consistency. Approximately 0.5 g of each homogenized sample was weighed accurately into digestion vessels, to which 2 mL of concentrated nitric acid (HNO₃) was added. Samples were allowed to predigest at room temperature for 2–3 hours, after which 1 mL of hydrogen peroxide (Hâ‚‚Oâ‚‚) was added to facilitate complete oxidation of organic matter. Microwave-assisted digestion was then carried out, gradually increasing the temperature to 200°C over 40 minutes, and maintaining at 200°C for an additional 30 minutes. After cooling to room temperature, digested samples were transferred to 50 mL volumetric flasks and diluted to volume with ultrapure water. The diluted samples were filtered through ashless filter paper (Whatman Grade 41) to remove any residual particulate matter. The resulting clear solutions were subsequently introduced into the calibrated ICP-OES instrument for mineral quantification. Calibration was performed using appropriate standards to ensure the accuracy and reproducibility of the measurements. |
|
C6. Line 364 In section 2.18. (Statistical Analysis) please provide information which software was used for statistical analysis of obtained data in the study. |
Line 389-391: Statistical analysis was carried out using Excel 2016 (Microsoft Corporation, Redmond, Washington, DC, USA) and Statistica 12 PL (StatSoft, Inc., Tulsa, OK, USA) software packages. |
|
C7. Line 373 – 374 The sentence “The study [35] found that a 9 kGy gamma dose completely eliminated Clostridium perfringens spores inoculated on ready-to-eat beef tripe.” should be changed to “The study of Parry-Hanson et al. [35], found that a 9 kGy gamma dose completely eliminated Clostridium perfringens spores inoculated on ready-to-eat beef tripe.” |
Line 414 Corrected |
|
C8. Line 413 I kindly ask the authors to use Italics for bacteria species. In sentence at line 413, Staphylococcus aureus should have been written in Italics, Staphylococcus aureus. Please check the entirety of the manuscript for similar mistakes. |
Corrected |
|
C9. Line 451 In table 4. Please explain the meaning of ±0.46. I believe that this represents standard deviation. However, this should be explained both in table and caption. |
Added Values were expressed as the mean ± SD |
|
C10. Line 576 Similar to in table 4., in table 5, please explain the meaning of ± symbol and provide explanation in both, table and caption.
|
Values were expressed as the mean ± SD |
Reviewer 2 Report
Comments and Suggestions for Authors
The manuscript explores the impact of electron-beam irradiation on various quality and safety parameters of beef. The study is timely and relevant, particularly due to growing global concerns about food safety, antimicrobial resistance, and food preservation.
Several similar studies on meat irradiation exist in the literature. The manuscript does not sufficiently highlight how this study offers a novel contribution (e.g., specific new insights into Kazakhstan meat products? Novel microstructural observations?).
Why were 3, 6, and 9 kGy chosen? Reference is made to FDA/WHO limits, but was any preliminary testing done to select these?. these doses should be addressed.
Did meat sit at refrigerated or room temperature during microbial testing? the storage condition post-irradiation should be clarified.
Describe meat sample uniformity (age, fat content, cut).
The TBARS analysis did not performed in this study, why? since this is crucial to see the oxidation level post-irradiation.
For amino acid degradation, it would be useful to clarify whether the changes are due to actual degradation or analytical limitations post-irradiation (e.g., oxidation leading to derivatization inefficiencies).
Some data (e.g., lysine increase at 3 and 6 kGy) is counterintuitive — potentially due to protein unfolding? This should be discussed.
Language requires extensive proofreading. Awkward phrasing and redundant language persist throughout. for example: “less densely packed muscle fibers and increased interstitial voids” → Consider: “Electron microscopy revealed disruption in myofibrillar structure, with increased interstitial spacing.”
No mention of economic feasibility, consumer acceptance, or regulatory implications (though discussed in introduction). consider to highlight this matter in the end of discussion/conclusion
Author Response
Reviewer 2
The manuscript explores the impact of electron-beam irradiation on various quality and safety parameters of beef. The study is timely and relevant, particularly due to growing global concerns about food safety, antimicrobial resistance, and food preservation.
|
Comment |
Answer |
|
Several similar studies on meat irradiation exist in the literature. The manuscript does not sufficiently highlight how this study offers a novel contribution (e.g., specific new insights into Kazakhstan meat products? Novel microstructural observations?). |
Line 98-104: This study offers a novel contribution by focusing on meat irradiation specifically for Kazakhstan—a region with a rapidly growing meat production sector yet limited research on advanced preservation techniques. It delivers specific insights on optimal irradiation dosage that eliminates pathogens while preserving nutritional quality, essential for developing regional food safety protocols that can enhance shelf life and ensure safety during logistics and production, in strict accordance with international and Kazakhstan’s regulatory authorities. |
|
Why were 3, 6, and 9 kGy chosen? Reference is made to FDA/WHO limits, but was any preliminary testing done to select these?. these doses should be addressed. |
Line 152-157: The choice of these particular doses (3, 6, and 9 kGy) was guided by regulatory standards and previous scientific literature, which identifies 3 kGy as a moderate dose effective for significantly reducing pathogens, 6 kGy as intermediate (addressing a wider spectrum of microorganisms including resistant vegetative cells), and 9 kGy as a higher but still practically relevant dose—approaching the upper limit recommended by international authorities for broader microbial inactivation including spores and viruses. The irradiation doses of 3, 6, and 9 kGy selected in this research are commonly utilized and well-established within the scientific literature and international regulatory guidelines for meat irradiation. |
|
Did meat sit at refrigerated or room temperature during microbial testing? the storage condition post-irradiation should be clarified. |
Line 177-180: After undergoing irradiation, the samples were transported immediately with crushed ice and then maintained at a temperature of approximately 4 ±â€¯1 °C through-out the microbial testing period. This refrigerated storage is critical to accurately assess the efficacy of the irradiation process in controlling microbial populations. |
|
Describe meat sample uniformity (age, fat content, cut). |
Line 112-115: The meat samples were obtained immediately after slaughtering of cows (2-2.5 years old) from the peasant farm “Mukinov” of Ernazar village, Beskaragai district, Abai region (Kazakhstan). The meat obtained after slaughter was cooled in refrigerat-ed chambers at 0–(+2) °C for 24 h. After cooling meat was vacuum-packed and frozen to -18 °C.
Line 138-141: Beef meat samples (Longissimus dorsi muscle) weighing approximately 150-200 grams each were individually sealed under vacuum conditions (40–60 mm Hg) in polyethylene bags measuring 20×25 cm (Figure 2). Average chemical composition - protein 25%, fat 15%, moisture 59%.
|
|
The TBARS analysis did not performed in this study, why? since this is crucial to see the oxidation level post-irradiation. |
In our current work, we focused on alternative indicators of meat quality and shelf-life, including the acid number, protein profile, amino acid composition, and microstructural observations. However, we acknowledge that TBARS analysis is crucial for assessing lipid oxidation. In future studies, we plan to incorporate TBARS and additional oxidative stress markers to provide a more comprehensive evaluation of post-irradiation oxidation. |
|
For amino acid degradation, it would be useful to clarify whether the changes are due to actual degradation or analytical limitations post-irradiation (e.g., oxidation leading to derivatization inefficiencies). |
Line 537-541: Radiation treatment of meat primarily affects sulfur-containing and aromatic amino acids through oxidation and deamination. Protein cross-linking occurs from free radical formation, altering texture. While higher doses progressively degrade amino acids, including essential ones, approved irradiation levels typically cause minimal nutritional impact. These changes are dose-dependent and influence meat's sensory qualities. |
|
Some data (e.g., lysine increase at 3 and 6 kGy, Lysine Tyrosine Phenylalanine Histidine Leucine+isoleucine) is counterintuitive — potentially due to protein unfolding? This should be discussed.
|
Line 510-515: The increase in lysine, tyrosine, phenylalanine, histidine, and leucine+isoleucine at 3 and/or 6 kGy may be explained by protein unfolding. Ionizing radiation induces partial denaturation, exposing previously buried amino acid residues, which enhances their detection during analysis. This likely reflects improved analytical recovery rather than a net increase in amino acid concentration, a phenomenon consistent with protein unfolding observed in irradiated meat. |
|
Language requires extensive proofreading. Awkward phrasing and redundant language persist throughout. for example: “less densely packed muscle fibers and increased interstitial voids” → Consider: “Electron microscopy revealed disruption in myofibrillar structure, with increased interstitial spacing.” |
Corrected Line 21-22 |
|
No mention of economic feasibility, consumer acceptance, or regulatory implications (though discussed in introduction). consider to highlight this matter in the end of discussion/conclusion |
Added Line 687-703: Economic Feasibility, Consumer Acceptance, and Regulatory Implications Economic feasibility is promising since irradiation reduces spoilage, extends shelf life, and lowers recall costs, which can offset the initial capital expenditure for modern irradiation facilities. Consumer acceptance, while historically cautious due to misconceptions regarding radioactivity and quality alterations, is expected to improve through targeted education and clear labeling—factors that further drive market competitiveness. In the Republic of Kazakhstan, irradiated meat is regulated by the EAEU technical regulations TR TC 034/2013 “On the Safety of Meat and Meat Products” and TR TC 021/2011 “On the Safety of Food Products,” as well as by the national standard ST RK GOST R 52529-2007, which defines identification methods for irradiation. Sanitary control is performed in compliance with the country’s regulations on radiation safety. Consequently, when these established normative requirements are met, the production and consumption of irradiated meat are considered acceptable and safe for consumers, as confirmed by the relevant regulatory documents. This robust framework, combined with economic and consumer factors, underscores the viability of adopting irradiation technology in Kazakhstan’s meat industry. |
Round 2
Reviewer 2 Report
Comments and Suggestions for Authors
The authors have revised their content